# SARS-CoV-2 Omicron (B.1.1.529) Infection of Wild White-Tailed Deer in New York City

**DOI:** 10.3390/v14122770

**Published:** 2022-12-12

**Authors:** Kurt J. Vandegrift, Michele Yon, Meera Surendran Nair, Abhinay Gontu, Santhamani Ramasamy, Saranya Amirthalingam, Sabarinath Neerukonda, Ruth H. Nissly, Shubhada K. Chothe, Padmaja Jakka, Lindsey LaBella, Nicole Levine, Sophie Rodriguez, Chen Chen, Veda Sheersh Boorla, Tod Stuber, Jason R. Boulanger, Nathan Kotschwar, Sarah Grimké Aucoin, Richard Simon, Katrina L. Toal, Randall J. Olsen, James J. Davis, Dashzeveg Bold, Natasha N. Gaudreault, Krishani Dinali Perera, Yunjeong Kim, Kyeong-Ok Chang, Costas D. Maranas, Juergen A. Richt, James M. Musser, Peter J. Hudson, Vivek Kapur, Suresh V. Kuchipudi

**Affiliations:** 1Department of Biology, The Pennsylvania State University, University Park, PA 16802, USA; 2The Center for Infectious Disease Dynamics, Huck Institutes of the Life Sciences, The Pennsylvania State University, University Park, PA 16802, USA; 3Animal Diagnostic Laboratory, Department of Veterinary and Biomedical Sciences, The Pennsylvania State University, University Park, PA 16802, USA; 4Department of Veterinary and Biomedical Sciences, The Pennsylvania State University, University Park, PA 16802, USA; 5United States Department of Health and Human Services, Silver Spring, MD 20993, USA; 6Department of Animal Science, The Pennsylvania State University, University Park, PA 16802, USA; 7Department of Chemical Engineering, The Pennsylvania State University, University Park, PA 16802, USA; 8National Veterinary Services Laboratories, Veterinary Services, U.S. Department of Agriculture, Ames, IA 50010, USA; 9White Buffalo, Inc., Chester, CT 06412, USA; 10City of New York Parks & Recreation, New York, NY 10029, USA; 11Laboratory of Molecular and Translational Human Infectious Disease Research, Center for Infectious Diseases, Department of Pathology and Genomic Medicine, Houston Methodist Research Institute and Houston Methodist Hospital, Houston, TX 77030, USA; 12Departments of Pathology and Laboratory Medicine and Microbiology and Immunology, Weill Cornell Medical College, New York, NY 10021, USA; 13Consortium for Advanced Science and Engineering, University of Chicago, Chicago, IL 60637, USA; 14Division of Data Science and Learning, Argonne National Laboratory, Argonne, IL 60439, USA; 15Department of Diagnostic Medicine/Pathobiology, Kansas State University, Manhattan, KS 66506, USA

**Keywords:** SARS-CoV-2, omicron, white-tailed deer, reinfection, reservoir competence, variant of concern, *Odocoileus virginianus*, spillover, zoonotic, disease ecology, enzootic transmission

## Abstract

There is mounting evidence of SARS-CoV-2 spillover from humans into many domestic, companion, and wild animal species. Research indicates that humans have infected white-tailed deer, and that deer-to-deer transmission has occurred, indicating that deer could be a wildlife reservoir and a source of novel SARS-CoV-2 variants. We examined the hypothesis that the Omicron variant is actively and asymptomatically infecting the free-ranging deer of New York City. Between December 2021 and February 2022, 155 deer on Staten Island, New York, were anesthetized and examined for gross abnormalities and illnesses. Paired nasopharyngeal swabs and blood samples were collected and analyzed for the presence of SARS-CoV-2 RNA and antibodies. Of 135 serum samples, 19 (14.1%) indicated SARS-CoV-2 exposure, and 11 reacted most strongly to the wild-type B.1 lineage. Of the 71 swabs, 8 were positive for SARS-CoV-2 RNA (4 Omicron and 4 Delta). Two of the animals had active infections and robust neutralizing antibodies, revealing evidence of reinfection or early seroconversion in deer. Variants of concern continue to circulate among and may reinfect US deer populations, and establish enzootic transmission cycles in the wild: this warrants a coordinated One Health response, to proactively surveil, identify, and curtail variants of concern before they can spill back into humans.

## 1. Introduction

Recent investigations have established that severe acute respiratory syndrome coronavirus-2 (SARS-CoV-2) infects a wide range of non-human animal hosts, including farmed mink, companion animals (e.g., cats, dogs, ferrets), and zoo animals (e.g., tigers, lions, cougars, snow leopards, gorillas, otters, and hippopotami) [1,2,3,4,5]. White-tailed deer (*Odocoileus virginianus*) are highly susceptible to SARS-CoV-2, as evidenced by experimental infection studies [6,7] and documentation of widespread natural infections in Iowa [8], Ohio [9], Pennsylvania [10], and several other States in the US, including New York [11]. Most recently, SARS-CoV-2 infections in deer from Canada have been confirmed, with evidence for long-term evolution within deer, and spillback to humans, further heightening concerns about the potential of deer to serve as a reservoir of SARS-CoV-2 [12].

Recent SARS-CoV-2 variants, such as Delta and Omicron, are more highly transmissible between humans than those previously described [13,14]; indeed, the effective reproduction number of Omicron in humans is estimated to be nearly threefold greater than the Delta variant [15]. While there are reports of Delta variant spillover into multiple animal hosts—including cats, dogs, pumas, and lions in a zoo in South Africa [16], as well as white-tailed deer across many US states, and Syrian hamsters in pet shops in Hong Kong [17]—spillover of the Omicron variant to non-human animal species has not yet been documented. There is a lack of clarity on the origins of Omicron, with competing hypotheses including emergence from a chronically infected human host, silent spread in a cryptic human population, or emergence from a yet-unknown non-human animal population [18].

While the widespread SARS-CoV-2 spillover infection of white-tailed deer across North America [8,9,10,11,12,19] has raised the possibility that deer could serve as a SARS-CoV-2 reservoir, there are several unanswered questions: firstly, whether or not Omicron has spilled over to free-living deer; secondly, while experimental studies show that deer infected with SARS-CoV-2 develop neutralizing antibodies [6,7], an open question remains as to whether deer, like humans, can be reinfected with SARS-CoV-2, even in the presence of neutralizing antibodies; thirdly, while white-tailed deer that were experimentally infected with the SARS-CoV-2 wild-type or the Alpha variant remained largely asymptomatic [6,7], it remains unclear whether free-ranging deer infected with the more recent SARS-CoV-2 variants (e.g, Delta or Omicron) exhibit discernible clinical signs. To address these three questions, we collected and tested nasal swabs and serum samples, and we provided a clinical examination of free-ranging white-tailed deer in Staten Island, New York, between December 2021 and February 2022. 

Here, we report SARS-CoV-2 Delta and Omicron variant spillover infection in the white-tailed deer population inhabiting Staten Island, which is a borough of New York City. To our knowledge, this is the first report of Omicron infection in a wildlife species. The results show that a majority of the SARS-CoV-2 seropositive deer in Staten Island exhibited strong serum reactivity to the wild-type SARS-CoV-2 B.1 lineage or the Alpha variant. Furthermore, robust levels of neutralizing antibodies were found in two deer that were positive for Delta variant RNA in their nasal passages, suggesting that deer, like humans [20], may be reinfected with SARS-CoV-2 or exhibit early seroconversion from ongoing Delta infections. Taken together, our studies suggest that white-tailed deer are emerging as a wildlife reservoir of SARS-CoV-2, and this highlights an urgent need to better assess the spillback risks and the evolutionary trajectories of SARS-CoV-2 variants in non-human animal reservoirs.

## 2. Materials and Methods

### 2.1. Samples

White-tailed deer were opportunistically darted and anesthetized by a veterinarian during an ongoing deer sterilization program implemented by the City of New York Parks & Recreation (NYC Parks). Once darted, the animals were tracked and processed at a nearby sampling site. The anesthetized animals were given ear tags, and were sampled and released. The GPS location of the sampling site for each individual was recorded, and the animal’s sex and age were determined. All captured deer were examined by a qualified veterinarian for gross abnormalities or illnesses; the examination included a standard temperature, pulse, and respiration reading and pulse oximeter readings using the lingual technique. Patient monitoring continued during the surgical process, and discontinued when the patient was again ambulatory. Blood samples were collected from the jugular vein, into serum separator tubes, and the serum was frozen at −20 °C until testing. Nasal swabs were collected by inserting Copan floQ swabs (Copan Diagnostics Inc.) into the nostril and touching the sides of the nasal wall. The swab was rotated for 30 s in each nostril, and was placed directly in Universal Transport Media (Copan Diagnostics Inc). The samples were submitted to the Penn State Animal Diagnostic Laboratory (ADL) for diagnostic testing, and were subsequently used in this study.

### 2.2. Surrogate Virus Neutralization Test (sVNT)

The serum samples were screened, using a surrogate virus neutralization test (sVNT) assay that had previously been validated for detecting SARS-CoV-2 antibodies in deer [21]. The sVNT assay uses cPass™ technology (Genscript), and detects total neutralizing antibodies measured as percent inhibition [22]. Animals with inhibition above 30% are considered positive.

### 2.3. Generation of SARS-CoV-2 S Pseudotyped Viruses

Pseudoviruses were produced, using a third-generation human immunodeficiency virus packaging system, as previously described [23]. Three plasmids—the transfer plasmid encoding luciferase and ZsGreen (BEI Resources Cat no: NR-52516), the helper plasmid encoding Gag/pol (BEI Resources Cat no: NR-52517), and the spike encoding plasmid of variants described in the study—were co-transfected in HEK 293T cells propagated in DMEM with 10% FBS, and maintained at 37 °C. Pseudovirus-containing supernatants were collected after 48 h, filtered through 0.45 μM low-protein binding filters, and were aliquoted and stored at −80 °C until further use.

### 2.4. Neutralization Assay of Deer Sera against Pseudotyped Virus Expressing the Spike Protein of SARS-CoV-2 Wild-Type or Variants

The deer serum samples that tested positive in the sVNT assay were examined for neutralizing activity with a SARS-CoV-2 pseudovirus neutralization assay (pVNT), using pseudotyped viruses that carried the S protein—which represented the wild-type SARS-CoV-2 B.1 lineage, Alpha, Beta, Gamma, Delta, or Omicron variants—to test the relative neutralizing titers against the SARS-CoV-2 variants. Pseudovirus neutralization assays were performed, using HEK 293T cells expressing ACE2, and TMPRSS2 cells (293T ACE2/TMPRSS2; BEI Resources Cat no: NR-55293), as described previously. Briefly, the pseudoviruses were incubated with threefold serial dilutions of sera for an hour at 37 °C. The pseudovirus/sera mixtures were subsequently inoculated into 96-well plates seeded with 3.0 × 10^4^ 293T ACE2/TMPRSS2 cells/well, a day before the assay. The residual pseudovirus infectivity was determined 48 h later, by quantifying the luciferase activity. The percentage neutralization was calculated, upon normalization to a virus-only control. Each serum was run in duplicate, in two independent experiments against each pseudovirus, to determine the 50% neutralization titer (NT_50_). The curves were fitted using a nonlinear regression curve, for which GraphPad Prism Software version 6 (San Diego, CA, USA) was employed; connected scatterplots were made using R software (R version 4.1.3).

### 2.5. Antigen Cartography

The antigen cartography was created using the NT_50_ measurements of the serum samples against the SARS-CoV-2 B.1 lineage, Alpha, Beta, Gamma, Delta, and Omicron variants in the pseudovirus neutralization assays. The distance between the SARS-CoV-2 (B.1 lineage and variants) and serum samples was calculated, and antigenic maps were generated, as described previously, using antigen cartography software accessed on 30 May 2022 (https://acmacs-web.antigenic-cartography.org/) [24,25]. The SARS-CoV-2 B.1 lineage, Alpha, Beta, Gamma, Delta, and Omicron variants, and the serum samples were positioned on a two-dimensional (2D) antigen map, based on the distances calculated by the antigen cartography algorithm, as described by Smith et al. [24]. The confidence area of the positions of the SARS-CoV-2 B.1 lineage, Alpha, Beta, Gamma, Delta, and Omicron variants, and the serum samples, were indicated as blobs, as estimated with stress parameter 0.1 [25]. The distance in antigenic units between the SARS-CoV-2 and serum samples was plotted, using GraphPad software version 9.0.0 (San Diego, CA, USA). A statistical analysis was performed, using a two-tailed unpaired Student’s t test with Welch’s correction. A *p* value of <0.05 indicated that the mean distance between the SARS-CoV-2 (B.1 lineage and variants) and the serum samples was significant.

### 2.6. RNA Extraction and RT-PCR for SARS-CoV-2 Detection

The swab samples were processed, and real-time RT-PCR was undertaken, following the standardized protocols for SARSR-CoV-2 detection in animal samples at Penn State’s ADL. RNA was extracted from 400 µL of swab samples, using a KingFisher Flex machine (ThermoFisher Scientific, Waltham, MA, USA) and a MagMAX Viral/Pathogen extraction kit (ThermoFisher Scientific, Waltham, MA, USA), following the manufacturer’s instructions. The presence of SARS-CoV-2 viral RNA was further tested, using the OPTI Medical SARS-CoV-2 RT-PCR kit, which is a highly sensitive assay that targets the N gene [26,27]. The RT-PCR assays were carried out on an ABI 7500 Fast instrument (ThermoFisher Scientific, Waltham, MA, USA). The internal control RNase P was utilized, to confirm that the samples were not contaminated with human tissue or fluids during harvesting or processing. The samples were also tested using a TaqPath kit (ThermoFisher Scientific, Waltham, MA, USA), which targets the SARS-CoV-2 ORF1ab, N gene, and S gene [27,28] as the first screen for Omicron, which typically presents as an S gene drop out in these assays [28].

### 2.7. SARS-CoV-2 Genome Sequencing

The total RNA extracted from the swab samples was used for the whole genome sequencing of the SARS-CoV-2, as previously described [8,29,30,31,32], and the sequencing libraries were prepared according to version 4.1 of the ARTIC nCoV-2019 protocol (https://artic.network/ncov-2019, accessed on 15 February 2022). We used a semi-automated workflow, which employed BioMek i7 liquid-handling workstations (Beckman Coulter Life Sciences) and MANTIS automated liquid handlers (FORMULATRIX). Using a NovaSeq 6000 instrument (Illumina), we generated short sequence reads, to ensure a very high depth of coverage. The sequencing libraries were prepared in duplicate, and were sequenced with an SP 300 cycle reagent kit.

### 2.8. SARS-CoV-2 Genome Sequence Analysis and Identification of Variants

The viral genomes were assembled, using the BV-BRC SARS-CoV-2 assembly service [32,33], which uses a pipeline that is similar to the One Codex SARS-CoV-2 variant-calling pipeline [34]. Briefly, the pipeline uses seqtk version 1.3-r116 for sequence trimming [35], minimap version 2.1 [36] for aligning the reads against the reference genome Wuhan-Hu-1 NC_045512.2 [36,37], samtools version 1.11 [35] for sequence and file manipulation [38], and iVar version 1.2.2 [39] for primer trimming and variant calling [40]. To increase stringency, the minimum read depth for the assemblies (based on samtools mpileup) was set at three, to determine consensus. Genetic lineages, variants being monitored, and variants were identified and designated by Pangolin version 3.1.11, with the pangoLEARN module 2021-08-024, using the previously described genome sequence analysis pipelines [29,32,41]. Single Nucleotide Polymorphisms (SNPs) were identified, using the vSNP (https://github.com/USDA-VS/vSNP, accessed on 6 March 2022) SNP analysis program.

### 2.9. Data Analysis and Visualization

QGIS mapping software version 3.16.10 was used, to visually portray the geographic location of the white-tailed deer that were sampled [38].

## 3. Results

### 3.1. Molecular and Genetic Identification of SARS-CoV-2 Delta and Omicron Variants in Nasal and Tonsillar Swabs from White-Tailed Deer on Staten Island, New York

Our results show that 8 out of 71 (11.3%, 95% CI: 0.0–0.20) white-tailed deer tested positive for SARS-CoV-2 RNA (Figure 1a). To determine the identity and genetic relatedness of the circulating strains, we applied whole-genome sequencing, using a recently described pipeline [8]. Our analysis confirmed that four of the SARS-CoV-2 PCR positive samples were the Delta variant, and that the other four positive samples were the Omicron variant. Notably, the Omicron variant was the dominant circulating lineage (90%) amongst humans in New York City during the period of the deer sampling, in late 2021 and early 2022 [42]. To our knowledge, this is the first report of the Omicron variant of SARS-CoV-2 infecting white-tailed deer or any other free-living wildlife.

We performed whole-genome-sequence-based phylogenetic analyses of these newly identified Omicron and Delta sequences, with the vSNP pipeline [43] (Figure 1b). The analysis showed that the Omicron sequences recovered from the deer were clustered closely with recently reported Omicron sequences from humans in New York City, as well as with those reported from environmental sources elsewhere, but were quite distinct from the previously described isolates recovered from deer in Iowa, Ohio [8,9], 14 other US states, and Canada, from which sequences had been deposited in GISAID [11] (Figure 1b and Appendix A). The Delta sequences were clustered closely with other deer Delta sequences that had been recently reported in multiple regions in North America (Figure 1b).

### 3.2. Delta and Omicron Infections Occur among Staten Island Deer despite Serological Evidence of Prior SARS-CoV-2 Exposure

It was not clear whether deer previously exposed to SARS-CoV-2 might be reinfected, and whether there was continued SARS-CoV-2 spillover infection of deer in urban settings after the emergence of the Omicron variant. To examine whether white-tailed deer on Staten Island had previously been exposed to SARS-CoV-2, we collected serum samples from 135 individual deer, between 12 December 2021 and 2 February 2022, and examined them for the presence of anti-SARS-CoV-2 neutralizing antibodies, using a surrogate virus neutralization assay (sVNT) [21] (Figure 2a). Due to the sampling design targeting males, most of the serum samples were from males (*n* = 119; 88.1%), with an age distribution skewed toward younger age classes. Eighty-six fawns constituted 63.7% of the sample, and 33 yearling deer made up 24.4%, while 16 of the 135 individuals (11.9%) were considered adults. Our sVNT results showed that 19 of the 135 (14.1% 95% CI: 0.0–0.20) serum samples were positive for SARS-CoV-2 exposure. Viral inhibition in the positive samples ranged from 33.2% to 97.0%, with a median value of 70.9% (Figure 2b and Appendix A). The proportion of positive animals was comparable to the findings of Chandler et al., who identified 9 out of 29 (31%; 95% CI 17%–49%) white-tailed deer, from two other New York counties, that were seropositive to SARS-CoV-2 in 2021 [21].

To determine which SARS-CoV-2 variant the deer were likely exposed to previously, we utilized a SARS-CoV-2 pseudovirus neutralization assay (pVNT) with six pseudotyped viruses equipped with the spike proteins of the wild-type SARS-CoV-2 B.1 lineage or that of the Alpha, Beta, Gamma, Delta or Omicron variants. In the pVNT assay, 11 out of the 19 (58%) sVNT positive samples showed stronger reactivity to the spike of the wild-type SARS-CoV-2 B.1 lineage than to any of the variants (Figure 2b), and one of the serum samples had comparable reactivity to all the variants except Omicron. Serum samples from seven of the deer (37%) reacted most strongly to the spike of Delta, while one deer reacted most strongly to the Alpha variant (Appendix A). Notably, none of the positive samples showed strong reactivity to the Omicron variant, compared with the wild-type B.1 lineage or other variants.

To investigate the antigenic relationship between the SARS-CoV-2 strains, we constructed antigen cartography, using the NT_50_ of serum samples determined by pVNT on the SARS-CoV-2 B.1 lineage, and on the Alpha, Beta, Gamma, Delta, and Omicron variants. The 2D antigenic map revealed clustering of 16 out of 19 sVNT positive samples to the B.1 lineage (antigenic units 0.26 to 0.89) and the Alpha variant (antigenic units 0.45 to 1.15), whereas 14 out of the 19 samples clustered with the Delta variant (antigenic unit < 1) (Figure 3 and Appendix A). Most of the serum samples which showed strong reactivity to the B.1 lineage in pVNT also had significant reactivity to the Alpha and Delta variants, and vice versa. This indicated the cross-neutralizing potential of the serum samples between the B.1 lineage, Alpha, and Delta variants, as they were close to the B.1 lineage, Alpha, and Delta variants in the antigenic map. The mean antigenic distances between the SARS-CoV-2 Gamma, Beta, and Omicron variants and the serum samples were 1.79 ± 0.74, 2.02 ± 0.74, and 3.2 ± 0.7, respectively, and they were significantly higher (*p* < 0.001) than the antigenic distance to the B.1 lineage (Figure 3b). Both the antigen map and the antigenic units indicated that the SARS-CoV-2 Omicron variant was distantly related to other strains (Figure 3 and Appendix A), as reported earlier [44]. As we used the sVNT with B.1 lineage RBD for the initial screening of the deer sera, we may have missed positive samples for Omicron antibodies.

### 3.3. Nasal Shedding of SARS-CoV-2 RNA in White-Tailed Deer with High Levels of Neutralizing Antibodies without Clinical Signs

Experimental SARS-CoV-2 infection in deer rapidly elicits a neutralizing antibody immune response [6,7], but it is not yet clear whether these antibodies prevent reinfection. While direct evidence for reinfection of experimentally or naturally SARS-CoV-2-infected white-tailed deer is lacking, our field-based studies show that two RT-PCR positive deer had relatively robust levels of neutralizing antibodies (79% and 94% inhibition, Appendix A). The pVNT assay of the serum sample from deer 2089, that was found to be infected with the Delta variant, showed relatively greater reactivity to the S protein of the wild-type B1 (50% more neutralizing titer (NT_50_ 1020) than Delta (NT_50_ 858) (Appendix A and Figure 2b)). Deer 2103 showed relatively higher reactivity to the Delta S protein (NT_50_ 1829) than to the B.1 S protein (NT_50_ 726). It is possible that either one or both individuals seroconverted during previous infection by an earlier SARS-CoV-2 lineage, and were reinfected with the Delta variant, a scenario that has also been observed in humans [20,45]. Alternatively, it is possible that these young individuals with high Ct values may have rapidly seroconverted in response to SARS-CoV-2 Delta infection, even while continuing to shed viral RNA in nasal secretions, as is observed in some SARS-CoV-2-positive humans [46] and in experimental infection of deer [6,7]. However, the small sample size and the modest differences in the pVNT titers do not conclusively support either reinfection or rapid seroconversion. Therefore, these competing hypotheses need to be rigorously tested, through experimental challenge studies and intensive field investigations that should include longitudinal sampling of individual animals and deer herds. It is noteworthy, however, that most of the deer in this study that were shedding viral RNA showed no detectable neutralizing antibodies. All four Omicron RNA-positive deer, and two of the four Delta RNA-positive individuals, tested negative for serum antibodies. The ability of SARS-CoV-2 variants to overcome neutralizing antibody responses to an earlier lineage/variant in deer is important, as it may imply that deer with antibodies to one variant could be susceptible to and transmit other variants. Such a scenario has the potential for a substantial impact on the overall transmission dynamics of SARS-CoV-2 in free-ranging deer, as well as their establishment as long-term reservoir hosts.

Previous studies have suggested that white-tailed deer remain largely asymptomatic following experimental SARS-CoV-2 infection [6,7]. Clinical examination of the deer in this study showed no gross abnormalities, including the eight deer who tested positive for Delta or Omicron RNA: as the exact day of infection cannot be known, it is also possible that these deer were past the disease stage of infection. However, if SARS-CoV-2 continues to cause low-pathogenicity infections in wild deer, then it could persist, unnoticed, within the deer population, which would facilitate the occurrence of viral evolution underneath our current surveillance radar.

## 4. Discussion

Multiple human-to-animal spillovers, and documentation of subsequent transmission among mink and deer, highlight the “generalist” and rapidly evolving nature of SARS-CoV-2 as a pathogen of mammalian hosts, as recently described [47]. Emerging evidence suggests adaption of SARS-CoV-2 in animal hosts: for example, six specific mutations in mink (NSP9_G37E, Spike_F486L, Spike_N501T, Spike_Y453F, ORF3a_T229I, and ORF3a_L219V), and one in deer (NSP3a_L1035F), have been identified [47]. Continued replication of SARS-CoV-2 in multiple animal hosts, and variation in selection pressure, could hasten viral evolution, and increase the likelihood of a novel strain emergence [48]. Notably, a recent manuscript reported a significant divergence between two deer-derived SARS-CoV-2 Alpha variant genomes that were also divergent compared to the human Alpha variant genomes from the same region, suggesting rapid viral evolution during deer-to-deer transmission [10,48]. Additionally, highly divergent SARS-CoV-2 sequences with 76 mutations, suggestive of host adaptation under neutral selection, have been identified in white-tailed deer in Canada [12]. Based on the identification of this highly divergent SARS-CoV-2, the authors also reported the first suspected white-tailed deer-to-human transmission [12]. Taken together, there is growing evidence that SARS-CoV-2 is likely to become established within deer populations, and that deer are likely to contribute to the emergence of novel variants of SARS-CoV-2. Considering the large population size and the potential for human–deer contact in certain environments, spillback of new strains of SARS-CoV-2 to humans and/or spread to other susceptible hosts is of genuine concern: hence, longitudinal studies to monitor SARS-CoV-2 adaptation and evolution within free-living deer and other susceptible non-human animal populations are urgently needed.

Several wild and synanthropic species—including raccoons, deer mice, and skunks—are susceptible to SARS-CoV-2 infection, and share ecological space with deer [49,50,51]; however, widespread SARS-CoV-2 spillovers into wild animals other than white-tailed deer (and perhaps mink) are currently unknown. Understanding the role of wildlife as potential reservoirs of new variants is essential, in order to assess the risk of future outbreaks in humans—particularly when the reservoir is peridomestic and abundant in urban areas, as is the case with white-tailed deer. While SARS-CoV-2 infections in white-tailed deer have been reported in 22 states and provinces in North America [11,12,19], a recent study reported that serological screening of roe, red, and fallow deer in Germany and Austria found no evidence of SARS-CoV-2 exposure in those animals, suggesting that differences in host susceptibility and ecological factors likely contribute to the potential for establishment of cervids as reservoir hosts [52]; however, the extent of SARS-CoV-2 infections in captive or free-living cervids (and indeed most other animal hosts) across the world remains poorly understood, and warrants investigation.

Examination of the virus receptor angiotensin-converting enzyme-2 receptor indicates that many mammalian families, including cervids and mustelids [3,47,53,54], are susceptible to SARS-CoV-2 infection, and that these models should be the focus of future sampling efforts. Viral infection of a wildlife host species does not necessitate that the host species sustain the virus within its population as a reservoir, as this depends on the ecological context, i.e., the complete ecosystem within which the species lives and interacts ecologically and epidemiologically [55]; therefore, a significant first challenge is to determine whether the temperate woodland community of North America (including white-tailed deer and other species) can sustain SARS-CoV-2 in the absence of continued spillover from humans. It is also possible that a sustained wildlife reservoir for SARS-CoV-2 could be replicated in, or generalizable to, other ecological communities around the world. Our studies highlight the urgent need for stratified surveillance of at-risk wild animal species, coupled with seasonal dynamical modeling, to predict the risk of SARS-CoV-2 spillback to humans.

## 5. Conclusions

Natural infections of wild white-tailed deer by the Delta and Omicron variants of SARS-CoV-2, in a herd previously exposed to SARS-CoV-2, emphasizes the role of white-tailed deer as a potential reservoir species. Establishing an animal reservoir might facilitate the continued circulation of SARS-CoV-2, independent of circulation in humans; in addition, deer might transmit the infection to other susceptible wild animals—such as rodents, foxes, and raccoons—resulting in the establishment of SARS-CoV-2 enzootic transmission cycles: such a scenario might result in virus adaptation, and the emergence of novel variants that might escape the protection of current SARS-CoV-2 vaccines. Therefore, these results are highly significant, and warrant the continued monitoring of white-tailed deer and other at-risk animal hosts for SARS-CoV-2 infections.

## Figures and Tables

**Figure 1 viruses-14-02770-f001:**
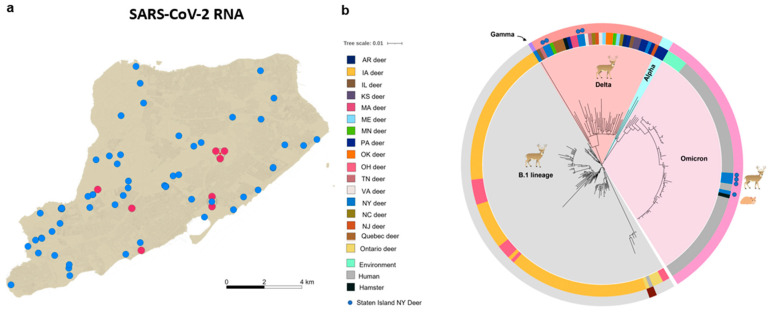
Distribution and whole-genome single nucleotide polymorphism (SNP)-based phylogenies of SARS-CoV-2 recovered from white-tailed deer on Staten Island, New York: (**a**) the spatial distribution of the collection sites of nasal and tonsillar swabs from white-tailed deer that were tested for the presence of SARS-CoV-2 viral RNA; red circles show sites where swabs were positive for SARS-CoV-2 viral RNA, and blue-filled circles show swabs that were negative; (**b**) whole-genome sequences of eight newly characterized white-tailed-deer-origin SARS-CoV-2 genomes were analyzed in the context of 135 publicly available white-tailed-deer-origin SARS-CoV-2 isolates, and 63 arbitrarily selected SARS-CoV-2 Omicron genomes circulating amongst humans in New York City during this same time period, as well as representative isolates from the environment or from Syrian hamsters (SI, Appendix A); the genome sequences were screened for quality, for SNP positions called against the SARS-CoV-2 reference genome (NC_045512), and for SNP alignments used to generate a maximum-likelihood phylogenetic tree, using RAxML.

**Figure 2 viruses-14-02770-f002:**
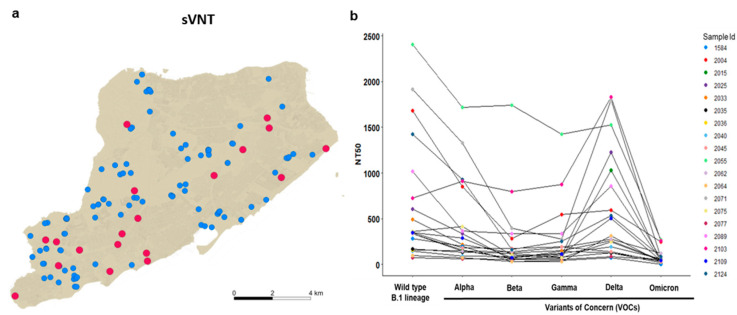
SARS-CoV-2 serological reactivity status of white-tailed deer on Staten Island, New York, between 12 December 2021 and 30 January 2022: (**a**) spatial distribution of sites of collection of serum samples from white-tailed deer for assessment of serological reactivity to, and neutralization of, SARS-CoV-2; red circles indicate positive detection, and blue-filled circles indicate seronegative status; (**b**) SARS-CoV-2 pseudovirus neutralization assay (pVNT) with pseudotyped viruses equipped with the spike proteins of the wild-type SARS-CoV-2 B.1 lineage, or that of the Alpha, Beta, Gamma, Delta or Omicron variants. Of the sVNT positive samples, 11 out of 19 showed stronger reactivity to the spike of the wild-type SARS-CoV-2 B.1 lineage than to any of the variants. Each serum was run in duplicate in two independent experiments against each pseudovirus, to determine the 50% neutralization titer (NT_50_). Connecting lines indicate serum from the same individual.

**Figure 3 viruses-14-02770-f003:**
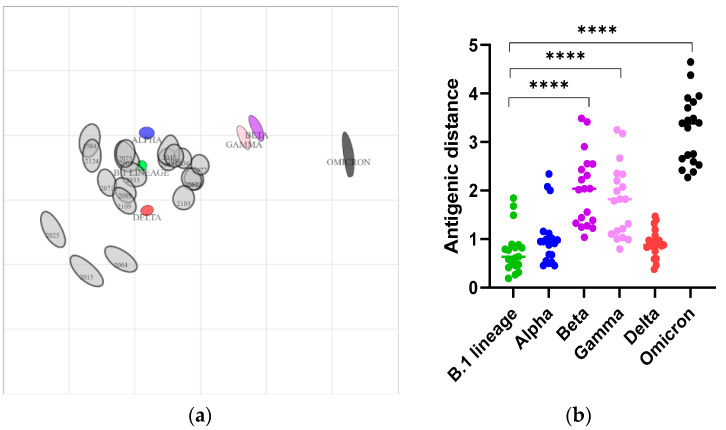
Construction of antigen cartography based on the NT_50_ of serum samples collected from deer. (**a)** Two-dimensional antigenic map of the SARS-CoV-2 B.1 lineage, the Alpha, Beta, Gamma, Delta, and Omicron variants, and the serum samples collected from the deer. The SARS-CoV-2 strains and serum samples are shown as blobs, as estimated using stress parameter 0.1. The SARS-CoV-2 B.1 lineage, and the Alpha, Beta, Gamma, Delta, and Omicron variants are indicated in green, blue, purple, pink, red, and black, respectively, and each gray blob corresponds to serum from one deer. Both the axes of the map are antigenic distant, and each grid square represents 1 antigenic unit, which is a three-fold serum dilution (two antigenic units correspond to nine-fold serum dilution, and so on) in the pseudovirus neutralization assay. The distance between points is a measure of antigenic similarity, with closer positions indicating higher antigenic similarity. (**b**) The distances of antigenic units between SARS-CoV-2 and the serum samples. The mean distance between the serum samples and the B.1 lineage, Alpha, and Delta variants was low, compared to the Beta, Gamma, and Omicron variants. *p* value < 0.05 are significant, **** *p* < 0.0001.

## Data Availability

All data used in this study are provided in Appendix A.

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
