# Peer review of "SARS-CoV-2 Omicron (B.1.1.529) Infection of Wild White-Tailed Deer in New York City"

_viruses, 2022, doi:10.3390/v14122770_

Round 1

Reviewer 1 Report

This is a very interesting study reporting spillback of SARS-CoV-2 to deer on Staten Island, New York.  The documentation of deer infected with the Omicron variant is important, as is the documentation of active infection in the presence of neutralizing antibodies.  The work is impressive, the laboratory analyses are very well done, and the quality of the writing is generally excellent.

I have two overarching comments.  First, a bit more ecological context would benefit this paper.  It will likely come as a surprise to certain readers that deer exist on Staten Island.  This is particularly the case for international readers who may not appreciate the ecological differences among New York City’s boroughs.  The authors should consider adding a section called “Study Sites” at the beginning of the Materials and Methods.  Here, they could briefly describe the habitat mosaic of Staten Island and the types of places that deer tend to inhabit.  This would help make clear to readers exactly how people and deer are interacting on Staten Island, which would help with the interpretation of the data.  Also, there is a literature on deer urban ecology on Staten Island, which could be described/cited here and in the Discussion.

Second, the Discussion section needs re-working.  As currently written, it contains no discussion of the specific results of this paper.  Rather, it is a generalized discussion of the state of knowledge about SARS-CoV-2 in wildlife (i.e. a literature review).  The authors have put a great deal of effort into characterizing viruses in Staten Island deer using molecular and serological tools, so they should begin the discussion with a detailed interpretation of their own data, including their implications and limitations.  Some of this can be addressed by moving certain sections from the Results to the Discussion.  As described below, there is a large amount of discussion-like text in the results sections of this paper.  However, additional text would be needed to link these points together into a cohesive section.

The following specific comments might also improve the manuscript.

Line 58.  Given that this study is about deer in New York, consider changing to “and several other states in the US, including New York[11].”

Lines 59-61.  Given that the papers are preprints, consider wording this sentence with a bit less certainty.  For instance, “Most recently a preprint claims to have documented SARS-CoV-2 infections in deer from Canada, including suggesting long-term evolution within deer and spillback to humans.”  The manuscript could very likely change during peer review.

Line 68.  There are several informal reports of Omicron spillback to zoo animals and deer, but none are yet published in the peer reviewed literature (of which I am aware).  Because of the rapidly changing situation, I suggest rewording this sentence as “spillover of the Omicron variant to non-human animal species has not yet, to our knowledge, been reported in the peer-reviewed literature.”

Lines 82-84.  Swabs and serum samples alone would not address the third question about clinical signs.  Add a phrase or sentence saying that clinical examinations were conducted.

Line 86.  International readers may not be familiar with Staten Island.  I suggest saying “Staten Island, a borough of New York City, New York, USA.”

Line 99.  It’s not clear what “opportunistically” means in this case.  If the deer were captured as part of a planned program to control deer populations, this may not be opportunistic.  Perhaps simply delete “opportunistically” here and elsewhere.  Using the word makes the sampling seem biased, but in fact it may not be significantly so.

Line 112.  Please provide a product ID and manufacturer for Universal Transport Media.  There are several commercial products with this name, and they differ in composition. I’m guessing this is the one from Copan, but it’s not clear.

Lines 161-163.  This appears to be an incomplete sentence.

Lines 201-206 and 209-210 and 228-232.  These sentences are largely background information and methods and are redundant with the information presented in the Introduction.  Consider starting with line 206 (“Our results show…”).

Lines 214-216.  I’ve heard rumors about similar findings elsewhere that are in various stages of publication.  I suggest removing statements of primacy such as this one.  Another paper documenting Omicron in wildlife could be published while this one is in press, for example.

Line 240.  Change “not different than” to “comparable to” because the values aren’t exactly the same.

Lines 285-297.  These interpretations are interesting, but they are not results so they should be moved to the Discussion.

Lines 298-303.  Consider moving these results to the top of the Results section.  That way, there is a natural flow from field to lab and from clinical signs to molecular and serological data.

Line 307 and 328.  Consider changing “Magenta” to “Red” (the dots look like a normal red color).

Figure 1.  It’s interesting that the terminal branches leading to Omicron variants in the phylogenetic tree are so much shorter than those leading to B.1 or Delta.  This is presumably because of the relative recency of Omicron’s spread.  It might be worthwhile to discuss this in the text.  Also, the drawing of the Syrian hamster next to the Omicron clade looks like a cat!

Line 352.  If the sample is from one deer, it should be “serum” (singular).

Figure 3 and Table S3. The purpose of this analysis isn’t clear from the text or figure legend. From Figure 3a, it would appear that deer are mounting immune responses to Alpha, B.1 and Delta, but not to Beta, Gamma or Omicron.  Figure 3b seems to be an alternative representation of the same data as in Figure 3a.  Is the purpose of the figure to show that deer are not reacting immunologically to Beta, Gamma or Omicron, or is this not the case?  Regardless, the authors should consider making clearer the purpose of this analysis and its interpretation for infected deer on Staten Island.

Lines 372-391.  The authors highlight specific mutations in SARS-CoV-2 in wildlife, including deer, but they do not say whether the sequences they generated in this study had any such SNPs.  The Staten Island deer strains closely cluster phylogenetically with known strains, but do they contain any novel mutations (nucleotide and amino acid)?  If there do, this would be a very important point to highlight. If there weren’t, this should also be presented and discussed.

Line 403 (and elsewhere).  Reference 54 is to a bioinformatics method published in 2009, but it is cited in support of cervids being a reservoir host.  There appear to be several references in the bibliography that, like this one, are cited in the wrong context.  The authors should go through their references and citations carefully to make sure they avoid such errors.

Author Response

Thank you very much for your timely and insightful comments! We have revised accordingly throughout the paper.

Reviewer 2 Report

This is a well written manuscript that clearly conveys the importance and need to continue evaluating SARS-CoV-2 infection in deer. The manuscript documents the first examples of Omicron in deer and is notable for establishing continued spillover of new variants into deer. 

The topic of re-infection by the two deer with neutralizing antibodies clearly addresses the potential these are just early conversions. However, the fact these deer had high Ct values and were young (1.5 and 0.5 years old) I think supports rapid seroconversion and not reinfection. I would highlight this and state reinfection is unlikely. The point it could indicate reinfection and that we should keep looking can remain.

Figure 1. is difficult to see and I recommend making two figures. I think the phylogeny is of interests and warrants a clear depiction. 

Author Response

Thank you very much for your timely and insightful comments! Yes, we tried mightily to temper our conclusions about reinfection because we do not have definitive evidence. We encapsulated this in our statement "the small sample size and the modest differences in the pVNT titers do not conclusively support either reinfection or rapid seroconversion." 

We have incorporated the suggested comments regarding age and high Ct values (ln 283). We have also updated both figures with higher quality images.

Reviewer 3 Report

Kurt J. Vandegrift et al., have an innovative and useful study to get clues for future pandemic or preparation to tackle them by understanding the possibility of zoonotic transmission or Deer to human transmission of COVID-19 and its variants.

Few things can be addressed:

-Why there is no ethical clearance number?

-Could you compare the trend in emergence of new variants being matched with Canadian study on Deer? Do you think they are prone to be infected by the newer strains than the wild type or the older strains of the SARS-CoV-2?

-While authors have highlighted the significance of the current work very well in the discussion and conclusion sections, abstract can be improved further to emphasize on this aspect.  

Author Response

Thank you for your comments.

Why there is no ethical clearance number?

-The samples used in this study were submitted (by White Buffalo Inc.) to the Penn State's Animal Diagnostic Lab for diagnostic testing. The Penn State Animal diagnostic laboratory has the necessary approvals for handling and processing the information.    

Could you compare the trend in emergence of new variants being matched with Canadian study on Deer?

-The Canadian study found a highly divergent SARS-CoV-2 variant. Our study found deer naturally infected with delta and omicron variants of SRS-CoV-2, genetically similar to the circulating variants in humans. Since our study is cross-sectional, we could not compare trends in variant emergence with the Canadian study. Longitudinal sampling in the exact locations could be helpful in comparing the trends in the emergence of variants.

Do you think they are prone to be infected by the newer strains than the wild type or the older strains of the SARS-CoV-2?

-It’s an excellent question. To answer this question, we first need to know the duration of the protective immune response in deer from natural infection. In addition, since the time and number of prior SARS-CoV-2 infections will be highly variable among free-living deer, it will be difficult to answer this question using field studies, and experimental infection studies can better address this question.

While authors have highlighted the significance of the current work very well in the discussion and conclusion sections, abstract can be improved further to emphasize on this aspect.  

-We revised the conclusion to emphasize the significance of this work, but we do not have much space to alter the abstract. Is there something specific that the reviewer would like to see in the abstract that is not there?  We will be happy to incorporate whatever they suggest.   

Reviewer 4 Report

The manuscript “SARS-CoV-2 Omicron (B.1.1.529) infection of wild white-tailed deer in New York City” provides a description of Omicron infection in wild white-tailed deer in Staten Island, New York. The authors have done genetic characterization from 8 SARS-CoV-2 RNA positive deer and found 19 animals seropositive to previous strains based on which the authors hypothesized the possibility of reinfection or early seroconversion. The authors have pointed out the supportive evidence to previous research and discussed the possibility of white-tailed deer as reservoir species and the potential of this to generate viruses that could then reinfect humans. The results do support the evidence of Omicron infection in white tailed deer but lack evidence of reinfection or early seroconversion. Yet the evidence of continued spill over infection in wildlife such as deer with different variants make this kind of study important. I have a few questions/comments below:

Major comments:

1. I think this manuscript should be rewritten as a short report. There are very few samples (even though it is hard to collect them), there are very limited analyses, and the display items can easily be presented in a single figure.

2. Line 116, which RBD(s) was/were used in surrogate virus neutralization test (sVNT) assay? It looks to me like the samples positive in this test were then 'typed' using pseudoviruses expressing spike from different VOC. But since B.1 and Omicron are antigenically very distant (at least in humans), you may miss Omicron-positive samples if the sVNT test is only performed with the B.1 RBD? This could explain why you have not found Omicron sero-positivity while Omicron appears to be prevalent in the deer.

3. Figure 1b for phylogenetic analysis, human and other specimens were used for comparison with Omicron sequences only. What is the difference between the genome sequence of Delta genome obtained from deer population compared to Delta sequence obtained from humans? Can Delta sequences from human population cases from the same area (if sequence/s are available) be included to compare it with deer Delta sequences obtained in this study? This could explain possible deer to deer transmission or human spillover infection to deer for Delta variants.

4. Line 236, it would be great to plot this as a function of age. It looks like almost all of the positive animals are 1.5 years old, with almost no adults or fawns being positive. This could give indications about longevity of response or susceptibility of age groups.

5. Line 239, live virus neutralization assay was tried but data not shown, how many samples were positive by live virus assay? Data should be shown, or this should be deleted.

Minor comments:

6. Line 38,155 deer were captured but there are only 135 sera and 71 swabs in this study. What determined whether these samples were collected or not?

7. Line 43, 'individuals' should be changed to 'animals'.

8. Line 43 'the first evidence' is a bit confusing and should be rephrased because it refers to two options (reinfection or early seroconversion). So technically you don't have evidence for either.

9. Line 85, authors report Delta and Omicron spillover infection into deer. Based on data from genetic characterization, the sequences obtained for Delta positive animals are closely related to sequences obtained from previous SARS-CoV-2 positive deer population so, is there a chance these were the result of deer-to-deer transmission?

10. Figure qualities are not good enough which make it difficult to understand the data e.g., the phylogenetic tree in Fig 1b is difficult to read, also the colors are difficult to identify in current resolution. Similarly in figure 2 and 3 the poor resolution makes it difficult to read the labels.

11. Line 275, please indicate animal ID here so these are easier to find in the table.

12. Reference 22 is incorrect in methods; it might be Tan C.W. et al. A SARS-CoV-2 surrogate virus neutralization test based on antibody-mediated blockage of ACE2–spike protein–protein interaction. Nat. Biotechnol. (2020).

13. Line 199, mentions nose and tonsillar swabs, but only results from nose swabs are listed. Please clarify if only nasal swabs were analyzed or both.

14. Line 203, the paragraph questions whether it is reinfection and continued spillover infection, the data from figure 1 do not suggest any possibility of reinfection, rather this question should be moved to result section 2 where serological evidence have been discussed.

15. Line 251, this could be due to a flaw in the experimental design if initial screening was not done with Omicron RBD (see above).

16. Line 299, “Clinical examination of the deer in this study showed no gross abnormalities, including the eight deer who tested positive”. The Ct values of all positive samples except 1 are high, which may indicate that the positive deer are past stage where disease signs could be observed. Should be added.

17. Line 308, the map here and in Fig. 2 is not useful without size/distance indicators.

18. It would be interesting to include some discussion on how deer acquired the infection, what is state of interaction between human and deer specific to this area.

Author Response

Thank you so much for your timely and insightful review.

Major comments:

  1. I think this manuscript should be rewritten as a short report. There are very few samples (even though it is hard to collect them), there are very limited analyses, and the display items can easily be presented in a single figure.

-This was the first detection of the Omicron variant infecting any wild animal in the world. It was also the first study to use live wild deer to investigate signs of disease with a veterinarian spending more than an hour monitoring each animal’s respiration. Finally, this is the first evidence of either re-infection or early seroconversion in wild deer which is critically important in assessing the likelihood these animals could be long-term reservoir species. We believe we will not be doing justice to all the details supporting our novel and critical findings in a short communication format.

  1. Line 116, which RBD(s) was/were used in surrogate virus neutralization test (sVNT) assay? It looks to me like the samples positive in this test were then 'typed' using pseudoviruses expressing spike from different VOC. But since B.1 and Omicron are antigenically very distant (at least in humans), you may miss Omicron-positive samples if the sVNT test is only performed with the B.1 RBD? This could explain why you have not found Omicron sero-positivity while Omicron appears to be prevalent in the deer.

-We did not see any Omicron seropositive deer in this study, mainly due to the recent introduction of this variant into the deer herds. We agree with the reviewer that we may miss Omicron-positive samples due to pre-screening all samples with the sVNT assay that uses B.1 RDB. However, in our study, we also screened several sVNT negative samples from this cohort for validating our pVNT assay, and we did not pick up any Omicron positive samples. While the B.1 and Omicron RBD are antigenically distinct, there is a fair amount of cross-reactivity between the two, so we could still find Omciron-positive sera if they were present, albeit with lower titers. 

  1. Figure 1b for phylogenetic analysis, human and other specimens were used for comparison with Omicron sequences only. What is the difference between the genome sequence of Delta genome obtained from deer population compared to Delta sequence obtained from humans? Can Delta sequences from human population cases from the same area (if sequence/s are available) be included to compare it with deer Delta sequences obtained in this study? This could explain possible deer to deer transmission or human spillover infection to deer for Delta variants.

-The focus of this paper was on Omicron as it was the novel finding.  Our group has a paper in PNAS that shows both deer-to-deer and human-to-deer transmission events are common. There have been several other papers that also confirm these findings in Ohio, Pennsylvania, and Canada.  

  1. Line 236, it would be great to plot this as a function of age. It looks like almost all of the positive animals are 1.5 years old, with almost no adults or fawns being positive. This could give indications about longevity of response or susceptibility of age groups.

-The deer in this study are part of a male sterilization project that has been ongoing for more than five years so only the new recruits into the population need sterilizations and thus were the only animals sampled. So, as we point out, there is a severe bias towards younger age classes. In addition, in the early stages of an emerging parasite invasion, age intensity and age prevalence analysis are not likely to be as telling as they are in long established parasite-host relationships as these patterns take time to settle in. Finally, age is a difficult thing to estimate with confidence in wild deer and so we can only be confident in three age categories (fawns, yearlings and adults) which are not enough categories for high quality age-class analyses.  The recent paper by Diego Diel’s group has a much larger sample size and shows clearly that, as expected, adult males are much more likely to harbor infections.

  1. Line 239, live virus neutralization assay was tried but data not shown, how many samples were positive by live virus assay? Data should be shown, or this should be deleted.

-The test was done as an internal control for the sVNT. We have deleted this statement from the text.

Minor comments:

  1. Line 38,155 deer were captured but there are only 135 sera and 71 swabs in this study. What determined whether these samples were collected or not?

-We joined this project opportunistically midway through its execution.

  1. Line 43, 'individuals' should be changed to 'animals'.

-This has been changed in the text.

  1. Line 43 'the first evidence' is a bit confusing and should be rephrased because it refers to two options (reinfection or early seroconversion). So technically you don't have evidence for either.

-I believe that technically we have the first evidence that one or the other is occurring and this is indeed novel. However, I have removed “the first” in the text.

  1. Line 85, authors report Delta and Omicron spillover infection into deer. Based on data from genetic characterization, the sequences obtained for Delta positive animals are closely related to sequences obtained from previous SARS-CoV-2 positive deer population so, is there a chance these were the result of deer-to-deer transmission?

- Deer-to-deer transmission is a crucial feature of SARS-CoV-2 infection of white-tailed deer, as shown by previous experimental infection studies. We believe that in natural human-to-deer spillover infection, few infected deer eventually spread the virus to other deer in the herd. So as the reviewer suggested, part of what we observed in this study could result from the deer-to-deer transmission. However, due to limited sampling, this is something we cannot definitively determine in cross-sectional field studies.

  1. Figure qualities are not good enough which make it difficult to understand the data e.g., the phylogenetic tree in Fig 1b is difficult to read, also the colors are difficult to identify in current resolution. Similarly in figure 2 and 3 the poor resolution makes it difficult to read the labels.

We have updated the draft with higher image quality figures.

  1. Line 275, please indicate animal ID here so these are easier to find in the table.

These deer numbers are provided on line 276 and 278, respectively.

  1. Reference 22 is incorrect in methods; it might be Tan C.W. et al. A SARS-CoV-2 surrogate virus neutralization test based on antibody-mediated blockage of ACE2–spike protein–protein interaction. Nat. Biotechnol. (2020).

We apologize for this confusion. We realized a technical issue with the Endnote and all of these have been fixed in that final updated version.

  1. Line 199, mentions nose and tonsillar swabs, but only results from nose swabs are listed. Please clarify if only nasal swabs were analyzed or both.

We analyzed both, but only found positive results in the nasal swabs.

  1. Line 203, the paragraph questions whether it is reinfection and continued spillover infection, the data from figure 1 do not suggest any possibility of reinfection, rather this question should be moved to result section 2 where serological evidence have been discussed.

We agree with this suggestion and the sentence has been moved to results section 2

  1. Line 251, this could be due to a flaw in the experimental design if initial screening was not done with Omicron RBD (see above).

We appreciate the valid technical question; we explained the justification for our experimental approach to the reviewer's previous comment (comment #2)

  1. Line 299, “Clinical examination of the deer in this study showed no gross abnormalities, including the eight deer who tested positive”. The Ct values of all positive samples except 1 are high, which may indicate that the positive deer are past stage where disease signs could be observed. Should be added.

Good catch! We have added a statement to reflect this on lines 302-303.

  1. Line 308, the map here and in Fig. 2 is not useful without size/distance indicators.

Another good catch! We have added a scale bar to each map in our new higher quality figures.

  1. It would be interesting to include some discussion on how deer acquired the infection, what is state of interaction between human and deer specific to this area.

This is New York City and the deer have very high contact rates with people and their byproducts.  They eat trash, lick salt off the sidewalks and breath the air. It is widely known that a great deal of hand feeding occurs. That said, this is a particularly difficult question to address definitively and is beyond the scope of our current work.

Reviewer 5 Report

It is well known that SARS co-2 is capable of infecting a wide range of animals and many animal species serve as a viral reservoir at the present thereby increasing the risk of zoonotic spillover and cross species transmission. Many of the know SARS-CoV2 variants are known to infect white tailed deer, but this is the first study showing the infection in white tailed deer with Omicron variant and spillover infection of Delta and omicron variants into the deer population in Staten, New York, highlighting the necessity to better assess and understand the risk of zoonotic transmission in animals. However, due to the small sample size, it is too early to make conclusions about omicron spillover, viral reinfection, disease symptoms in the infected animals and seroconversion. Below are my comments:

 Minor comments:

1)    In the method section describing the neutralization assay of deer sera against pseudovirus, the given reference (23) does not support the statement (Line 135-136). Please provide the correct reference.

2)    Line 167, reference 27, please correct

3)    Method section reference 27, 28, line 172, please correct.

4)    Reference 29, line 172, please correct.

5)    It is advised to provide high quality images in the manuscript as the current version of the images is a little blurry.

6)    Sequencing pipeline described to identify variants in the method section (ref 3,33, 43) does not match with the one provided in the result section (ref 45, line 218)

Major:

1)    The authors showed the occurrence of Delta and Omicron infections among Staten Island deer despite serological evidence of prior SARS-CoV-2 exposure but did not show any evidence of spillover and reinfection for the omicron variants. Please explain.

Author Response

 Minor comments:

1)    In the method section describing the neutralization assay of deer sera against pseudovirus, the given reference (23) does not support the statement (Line 135-136). Please provide the correct reference.

2)    Line 167, reference 27, please correct

3)    Method section reference 27, 28, line 172, please correct.

4)    Reference 29, line 172, please correct.

5)    It is advised to provide high quality images in the manuscript as the current version of the images is a little blurry.

6)    Sequencing pipeline described to identify variants in the method section (ref 3,33, 43) does not match with the one provided in the result section (ref 45, line 218)

-We had an issue with the reference software. All of the reference issues you raised have been corrected. We are also uploading higher quality images in this resubmission. 

The authors showed the occurrence of Delta and Omicron infections among Staten Island deer despite serological evidence of prior SARS-CoV-2 exposure but did not show any evidence of spillover and reinfection for the omicron variants. Please explain.

-We want to clarify that we did find Delta and Omicron variant spillover infection of white-tailed Deer in Staten Island. Further, our phylogenetic analysis based on the whole genome sequencing supports the spillover infection as the sequences were similar to the circulating Delta and Omicron variants in humans. In addition, our serology data support that the deer in State Island have previously been infected, as evidenced by neutralizing antibodies that strongly reacted to ancestral SASR-CoV-2 spike. Therefore, our data support the possibility that deer in Staten Island were previously infected with ancestral SARS-CoV-2 but were reinfected with the more recent Delta and Omicron variants similar to humans. However, we recognize that the small sample size in our study means that further evidence is needed to confirm the claim of reinfection of deer.

Thank you for your time and insightful comments!

Round 2

Reviewer 4 Report

1. The authors have replied to the concerns raised in the rebuttal letter, but many of them have not been rectified within the manuscript. As the concerns were raised from the reader’s point of view, clarity in the manuscript is needed.

2. The author states that “we also screened several sVNT negative samples from this cohort for validating our pVNT assay,” but nowhere is it mentioned in the manuscript. There is always the chance of missing Omicron positive samples by sVNT assay, as RBD is against B1, and according to authors, only a few samples were tested by pVNT assay, so there is the chance of missing the omicron-positive samples. The response by the authors that ‘we could still find Omicron-positive sera if they were present’ is not based on scientific evidence since the sera were never properly tested for the presence of Omicron antibodies. Thus, they cannot state there was no seropositivity for Omicron in these deer.

3. The scale bar for the map is still not included in the manuscript.

Author Response

Thank you again for your excellent comments!

Apologies, I added a scale bar and did upload them as much higher-quality zipped figures (1 & 2), but I did not insert them into the word document.  They are inserted into the document now. 

With respect to your second comment concerning the B.1 RBD of the sVNT, we agree with you (the reviewer); we may have missed some Omcron-positive samples as the initial screening was done with sVNT with B.1 RBD. We have inserted the following sentence in the manuscript (Line 267):

As we used sVNT with B.1 lineage RBD for the initial screening of deer sera, we may have missed positive samples for Omicron antibodies.